# Synthetic Data Generation with Lorenzetti for Time Series Anomaly Detection in High-Energy Physics Calorimeters

Laura Boggia[1,2] ⋆ and Bogdan Malaescu[2]

**1** IBM Research Paris-Saclay
**2** LPNHE, Sorbonne Université, Université Paris Cité, CNRS/IN2P3

⋆ laura.boggia@cern.ch

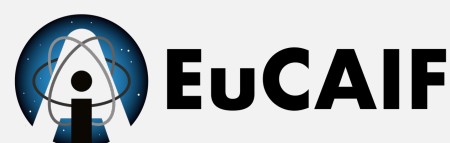

*The 2nd European AI for Fundamental Physics Conference (EuCAIFCon2025) Cagliari, Sardinia, 16-20 June 2025*

## Abstract

**Anomaly detection in multivariate time series is crucial to ensure the quality of data coming from a physics experiment. Accurately identifying the moments when unexpected errors or defects occur is essential, yet challenging due to scarce labels, unknown anomaly types, and complex correlations across dimensions. To address the scarcity and unreliability of labelled data, we use the `Lorenzetti` Simulator to generate synthetic events with injected calorimeter anomalies. We then assess the sensitivity of several time series anomaly detection methods, including transformer-based and other deep learning models. The approach employed here is generic and applicable to different detector designs and defects.**

## 1 Introduction

The timely detection of detector issues is an important challenge for Data Quality Monitoring (DQM) at High Energy Physics (HEP) experiments and requires substantial human input. The application of Time Series Anomaly Detection (TSAD) techniques to the problem might relieve some of this workload. Furthermore, many datasets used for TSAD suffer from unreliable or scarce labels. We propose injecting artificial detector defects in the calorimeter simulated by the `Lorenzetti` Simulator [1]. The synthetic defects allow us to systematically compare the effectiveness and sensitivity of various anomaly detection methods based on deep learning.

Deep learning–based TSAD methods have only recently been applied to fundamental physics, as many phenomena are time-invariant or lack precise timing due to detector effects. At the LHC, collisions are considered permutation-invariant, but detector readout introduces temporal dependencies. This makes TSAD promising for DQM, a key task for the experiments, and is especially relevant with the upcoming upgrade of the LHC that will lead to a significant increase in the data-taking rate. Recent work on DQM with ATLAS [2] includes a

TSAD approach for ATLAS data acquisition [3] and an LSTM-autoencoder for the ATLAS LAr calorimeter [4]. CMS [5] has investigated various supervised and unsupervised approaches to DQM [6–8], developed a CNN autoencoder for CMS electromagnetic calorimeter [9], and implemented an automated DQM approached using statistical and machine learning tests [10].

## 2 Synthetic Data Production

### 2.1 `Lorenzetti` **Showers**

`Lorenzetti` Showers is a calorimeter simulation framework for HEP [1], with default granularity based on the ATLAS experiment [2]. It provides calorimeter information including pileup and crosstalk, but no tracking, using `PYTHIA8` [11,12] for event generation and `Geant4` [13] for detector simulation. The simulation framework is subject to limitations stemming from computational efficiency and design decisions made in earlier implementations. First, in the detector simulation step, the default setup discards all cells not associated with the seed of a simulated particle and thereby drastically reduces the amount of saved information. Therefore, these cells are not accessible in later simulation stages and cannot be modified. Second, the simulation initially focused on the simulation of truth-level events, and there are still remnants of this in the current implementation. This hinders the random insertion of noise bursts in various regions of the simulated detector. Ongoing work aims at addressing this.

### 2.2 Dataset with Synthetic Detector Defects

The goal of this work is to introduce artificial, time-dependent detector defects into the simulation and to re-identify them using various TSAD approaches, to quantify their anomaly detection performance. The synthetic anomalies studied in this project are restricted to detector defects. They are introduced during the digitisation step, where the electronic signals, induced by the energy deposits in the calorimeter cells, are modelled and processed.

Concretely, two types of synthetic detector issues are implemented, namely anomalies due to **inactive calorimeter modules** and anomalies related to **increased noise** in the cells recording a physics signal. We carefully select varying detector regions and identify cells within those regions that will be affected by the artificial anomalies.

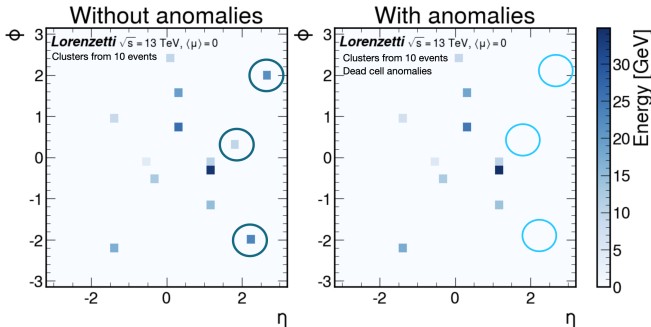

Figure 1: Phase $(\eta, \phi)$ space shows cell energies after digitisation, aggregated over 10 events. The left plot depicts the default simulation, while the right highlights anomalies from inactive modules (missing deposits in cyan circles).

**Dead (Inactive) Modules**    In the case of dead detector modules, the electronic signals from entire calorimeter regions are suppressed. This extreme anomaly is reliably flagged in DQM,

making it a suitable test case for validating TSAD algorithms. The implementation is straightforward: the electronic pulse is set to zero for all affected cells. In Figure 1, a visual example of such a defect, affecting 10 sequential collision events, is given. Only some detector regions on the right-hand side of the detector (i.e. $\eta > 0$) are concerned.

**Increased Noise**    In the second case, the electronic signal from the calorimeter cells is distorted more strongly than by the default noise. Initially, the noise is drawn from a Gaussian distribution with a default mean and standard deviation. To disturb this, the standard deviation for specific calorimeter cells is multiplied by an adjustable noise factor, giving rise to modified detector signals. This approach is limited to cells already containing a physics signal due to the limitations of `Lorenzetti`.

## 2.3   Data Preparation and Model Training

We simulate roughly 100k jet events with `Lorenzetti`, using the first 64k for training and injecting anomalies into the remaining 35k. For each event, we use the cluster energy $E$, transverse energy $E_T$, pseudorapidity $\eta$, azimuthal angle $\phi$, and the four shower shapes $r_\eta, r_\phi, r_{had}$, and $E_{ratio}$. To account for varying cluster multiplicity, features are aggregated via mean and standard deviation, then normalised to $[0, 1]$. In total, this amounts to $N = 18$ features.

   We select an unsupervised baseline, the reconstruction-based `iTransformer` (a variation of the forecasting-based `iTransformer` [14]), the `TranAD` [15] and the `USAD` [16] model and apply them to the synthetic `Lorenzetti` datasets. The unsupervised baseline applies the absolute value function to the time series. All models provide multidimensional outputs (called anomaly scores) that are converted into binary anomaly labels with the Peak-over-Threshold (POT) [17] method. Three strategies are considered:

- **global**: The anomaly scores are first averaged over all $N$ variates to get a univariate series of anomaly scores to which POT can be applied.

- **local (inclusive OR)**: POT is applied separately to each variate of the time series. Next, an *inclusive OR*-pooling function acts on the $N$ variates, i.e. if a time stamp is anomalous in at least one of the $N$ dimensions, the time stamp is anomalous in the final labels.

- **local (majority voting)**: Again, POT is applied separately to each variate of the time series and *majority voting* across the variates, i.e. if at least $N/2$ dimensions are anomalous at a certain time stamp, the time stamp is declared anomalous in the combined labels.

   Training runs for ten epochs and is repeated five times with different validation splits, using Mean Squared Error as the loss. A sliding window of 10 is applied across all models, with step size 1 for `iTransformer` and `TranAD`, and 5 for `USAD`. Further details are given in [18].

## 3   Results

The chosen TSAD models were tested on several datasets featuring artificial defects under different pileup scenarios, and their anomaly detection ability was assessed using the Matthew's Correlation Coefficient (MCC) score [19]. For each pileup level (i.e. $\langle \mu \rangle = 0, 40$), training was performed on data subjected to the same pileup conditions.

   As expected, the anomalies associated with inactive modules are rather easy to identify for the TSAD models, since they affect large phase space regions. All deep learning models seem to learn and correctly detect part of the anomalies (MCC scores range between $0.5 - 0.77$ depending on the model), while the baseline fails at doing so (MCC score of 0.006).

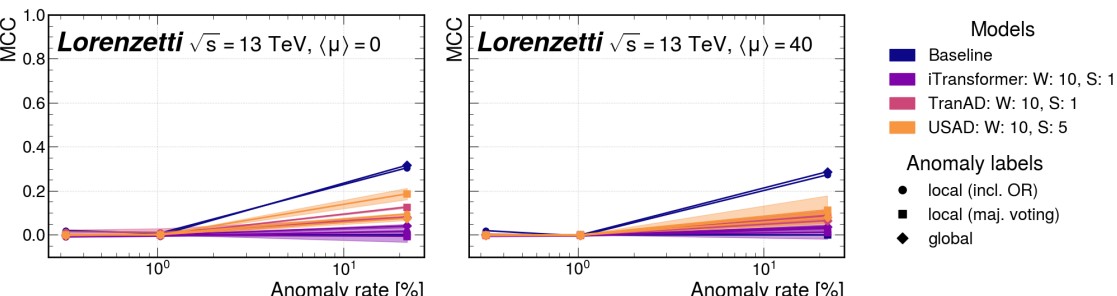

Figure 2: MCC scores (averaged over five random seeds) as a function of the anomaly rate in the test dataset without pileup (left) and with pileup (right) for three anomaly label methods.

All models struggle extremely to identify synthetic anomalies associated with increased noise, especially for low anomaly rates. Figure 2 shows the averaged MCC scores for each TSAD model as a function of the anomaly rate with and without pileup. The left plot presents the case without pileup, while the right plot displays results for the test data containing pileup at $\langle \mu \rangle = 40$. Results are similar to the no-pileup case, except that several models show larger MCC score variations with pileup. Only for the highest anomaly rate (i.e. 22%), the baseline manages to correctly identify some anomalies. Similarly, the USAD model does not fail completely, as the anomaly rate increases, and correctly tags some of the anomalous events. The low anomaly detection performance may stem from anomalies always appearing with real physics signals, violating the assumption that anomalies originate from a distinct distribution.

## 4  Conclusion

We presented the injection of artificial detector defects into the general-purpose calorimeter simulator `Lorenzetti`, focusing on two cases: added noise convoluted with physics signals and inactive detector modules. Two training datasets (with and without pileup) and multiple test sets with varied anomaly types and rates were produced.

Four TSAD models – an unsupervised baseline, `iTransformer`, `TranAD`, and `USAD` – were evaluated. As expected, all models rather successfully detected inactive modules, validating their ability to identify obvious defects. In contrast, anomalies from increased noise proved much harder to detect, partly due to limitations in the simulator, which cannot reproduce realistic coherent noise patterns uncorrelated from physics signals.

Future work should investigate the discriminative power of the shower shape variables, which showed some sensitivity to anomalies, and expand the anomaly simulation framework in `Lorenzetti` to enable broader studies of detector defects.

## Acknowledgements

Huge thanks to the `Lorenzetti` team – in particular to E.E.P. Souza, J.L. Marin, E.F. Simas Filho, L. Nunes and B. Laforge – for their technical support and discussions!

**Funding information**    This work is part of the SMARTHEP network and is funded by the European Union's Horizon 2020 research and innovation program, call H2020-MSCA-ITN-2020, under Grant Agreement n. 956086 (L.B.).

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
