# Peer review of "Synthetic Data Generation with Lorenzetti for Time Series Anomaly Detection in High-Energy Physics Calorimeters"

_SciPost Physics Proceedings_

## Round 1 · Referee Report · Anonymous (Referee 1) · 2025-10-31

Disclosure of Generative AI use

The referee discloses that the following generative AI tools have been used in the preparation of this report:

GPT-5 (mini) used to help with formatting and polishing the report and the requested changes

Report

The submission motivates the application of time-series anomaly detection methods on synthetic high-energy physics (HEP) data. Such techniques could significantly aid data quality monitoring in particle physics experiments by identifying new types of detector issues and reducing the human expert workload.

Different deep learning approaches are investigated and compared to a simple baseline. Various anomaly labeling strategies are explored to handle anomalies in multidimensional outputs. The study also examines the impact of different numbers of overlapping proton-proton collisions (pileup) on model performance.

The manuscript is transparent about its limitations, particularly that some anomalies correlated with the physics signal remain challenging for the tested methods. These challenges are primarily due to constraints of the underlying simulation framework rather than the methods themselves. The study should therefore be seen as a proof-of-concept.

Overall, it is a well-written and timely submission.

Requested changes

Content: 1- Introduction: cite other relevant recent literature, e.g., arXiv:2501.13789. 2- Section 2.1 (Lorenzetti showers): include a citation for the ATLAS experiment; clarify how the limitations in the framework affect the final time series data (e.g., for the truth-level simulation in the penultimate sentence). 3- Section 2.3: clarify the dimensionality (N) of the final time series. 4- Results: briefly discuss or speculate why the deep learning–based approaches struggle compared to the simple baseline.

Minor / Formatting: 5- Ensure the abstract is a single block of text without line breaks. 6- Explain technical terms where possible (pileup, ESD, TranAD, USAD, MCC), or otherwise minimize their use. 7- Dataset sizes: clarify the mismatch (64k training jets + 35k testing jets ≠ 100k). 8- References: provide a report number for Ref. 4 and DOI/arXiv information for Ref. 8. 9- Conclusion formatting: ensure the “Lorenzetti” text fits within the page margins.

Recommendation

Ask for minor revision

  • validity: high
  • significance: good
  • originality: good
  • clarity: high
  • formatting: excellent
  • grammar: excellent

Author:  Laura Boggia  on 2025-11-01  [id 5980]

(in reply to Report 1 on 2025-10-31)
Category:
remark

Hello,
thank you very much for your valuable comments!
They were very useful and I now implemented these changes as well as possible while respecting the length limitations. The revised version will appear on arXiv in the next few days and I'll then re-submit it here on SciPost.
Best regards,
Laura

---

## Round 2 · Referee Report · Anonymous (Referee 1) · 2025-11-17

Report

The authors have improved the submission and addressed most of the feedback from the first report (v1). I have just a few editorial suggestions for v2

Requested changes

Please double check the references: 1- Ref [8] appears to be a doctoral dissertation and should be stated as such. Please also add a bibliographic identifier or a link if possible 2- For some references the arXiv identifier appears twice. Sometimes the formatting is inconsistent (with/without prefix) 3- Ref [19] seems to have a broken link

Recommendation

Ask for minor revision

---

## Round 2 · Author Response

Hello,
This is the re-suvmitted version, taking into account the minor revision comments we received.
Thanks

---

## Round 3 · Author Response

Hello,
Thank you for your recommandations and please accept my apologies for the late resubmission.
I hope this new version addressed all the issues raised in the last comments (namely the format and coherence of the references).

---

## Editorial Decision

accepted_in_target_journal